# Cisplatin Dependent Secretion of Immunomodulatory High Mobility Group Box 1 (HMGB1) Protein from Lung Cancer Cells

**DOI:** 10.3390/biom13091335

**Published:** 2023-08-31

**Authors:** Kevin P. Gillespie, Ross Pirnie, Clementina Mesaros, Ian A. Blair

**Affiliations:** Center of Excellence in Environmental Toxicology, Department of Systems Pharmacology and Translational Therapeutics, University of Pennsylvania, Philadelphia, PA 19104, USA

**Keywords:** platinum drugs, acetylation, oxidation, immune response, mass spectrometry, chromatography

## Abstract

High mobility group box 1 (HMGB1) is secreted from activated immune cells, necrotic cells, and certain cancers. Previous studies have reported that different patterns of post-translational modification, particularly acetylation and oxidation, mediate HMGB1 release and confer distinct extracellular HMGB1 signaling activity. Here we report that cisplatin but not carboplatin induces secretion of HMGB1 from human A549 non-small cell lung cancer (NSCLC) cells. Cisplatin-mediated HMGB1 secretion was dose-dependent and was regulated by nuclear exportin 1 (XPO1) also known as chromosomal maintenance 1 (CRM1) rather than adenosine diphosphate (ADP)-ribosylation, acetylation, or oxidation. HMGB1, as well as lysine acetylation and cysteine disulfide oxidation of secreted HMGB1, were monitored by sensitive and specific assays using immunoprecipitation, stable isotope dilution, differential alkylation, and nano liquid chromatography parallel reaction monitoring/high-resolution mass spectrometry (nano-LC-PRM/HRMS). A major fraction of the HMGB1 secreted by low-dose cisplatin treatment of A549 NSCLC cells was found to be in the fully reduced form. In contrast, mainly oxidized forms of HMGB1 were secreted by dimethyl sulfoxide (DMSO)-mediated apoptosis. These findings suggest that inhibition of XPO1 could potentiate the anti-tumor activity of cisplatin by increasing the nuclear accumulation of HMGB1 protein, an inhibitor of cisplatin DNA-adduct repair. Furthermore, low-dose cisplatin therapy could modulate the immune response in NSCLC through the established chemokine activity of extracellular reduced HMGB1. This could potentially enhance the efficacy of subsequent immunotherapy treatment.

## 1. Introduction

HMGB1 was first isolated among other “high-mobility group” proteins in 1973, so designated because of their high mobilities in polyacrylamide gel electrophoresis relative to other extracted non-histone chromosomal proteins [1]. It is a 30 kDa protein with two nuclear localization signals (NLSs) and an acidic tail of 30 amino acids (Figure 1). HMGB1 interacts with DNA and other nuclear proteins to regulate replication [2], translation [3], and DNA repair [4]. Beyond its nuclear function, HMGB1 has a dual second role as a danger-associated molecular pattern (DAMP) by mediating an immune response when secreted from activated immune cells [5] or passively secreted from necrotic cells [6]. Secreted HMGB1 can activate the receptor for advanced glycation end products (RAGE) [7] as well as toll-like receptor (TLR)-2 and TLR-4 [8]. Among other consequences, activation of these receptors initiates cell-signaling cascades that activate nuclear factor kappa B (NF-κB), a key transcription factor in both innate and adaptive immune responses [8]. The mechanisms that regulate the shift between HMGB1′s dual nuclear and DAMP activities and their impact in human disease have been the subject of much of the research into HMGB1 as a potential therapeutic target [9,10].

The role of HMGB1 in the etiology of cancer is complex because of its involvement in both DNA repair and extracellular signaling [11,12]. Growth and metastasis of tumors is thought to be sustained by signaling from HMGB1 and other DAMPs released by necrotic cells following chemotherapy or radiation [13,14]. Extracellular HMGB1 promotes invasion and metastasis by stimulating cell migration and by mediating interactions among components of the extracellular matrix and the tumor microenvironment. Due to these functions, circulating HMGB1 is considered to be an indicator of tumor progression in many cancers [reviewed in [15]]. However, intracellular HMGB1 also has effects on tumor progression by mediating DNA repair mechanisms [4] and NF-κB signaling [8]. Although HMGB1 overexpression has been reported to be common in many types of cancer, including non-small cell lung cancer (NSCLC) [16,17], breast cancer [18], colorectal cancer [19], pancreatic cancer [20], and malignant mesothelioma [21], the ability of different types of tumors to produce their own pool of circulating HMGB1 remains uncertain. In addition, increased tumor HMGB1 mRNA and protein expression correlates with the progression of NSCLC patients [22]. The mechanisms that regulate HMGB1 localization to the nucleus, cytosol, and extracellular space are unclear and may vary among different contexts and cell types.

Platinum-based chemotherapies such as cisplatin and carboplatin are important therapeutic options for the treatment of NSCLC [23,24]. The cytotoxicity of platinum drugs in lung cancer cells is thought to result from DNA damage, especially 1,2-d(GpG) intrastrand crosslinks, which interferes with DNA replication and eventually activates apoptosis [25]. HMGB1 secretion during the treatment of NSCLC has been reported, although the studies were flawed by the analysis of HMGB1 in serum instead of plasma [26,27]. Regrettably, a significant number of studies of HMGB1 secretion have been conducted using serum instead of plasma [28] Our study utilizing matched plasma and serum samples from twenty healthy control subjects revealed that HMGB1 is released when blood is allowed to clot and form serum [15]. The process of clotting led to a substantial 30-fold increase in HMGB1 concentrations, escalating from 0.2 ng/mL in plasma to 6 ng/mL in serum [15]. Consequently, the reliable detection of an elevation in HMGB1 release from tissues into the plasma becomes very difficult to interpret when the protein is quantified in serum. Measurements conducted solely in serum only indicate the ability of blood cells to generate HMGB1 and fail to reflect the actual circulating levels of HMGB1. Despite this fundamental issue, serum is still frequently employed in biomarker studies to quantify HMGB1 secreted from tissues in various pathological conditions. 

Various post-translational modifications (PTMs) of HMGB1 have been implicated as mediators of HMGB1 mobility and secretion. These modifications include acetylation [29,30,31,32] methylation [33], poly-ADP ribosylation [34], serine phosphorylation [30,31], and cysteine (Cys) oxidation [31]. A paradigm has emerged relating various circulating HMGB1 proteoforms with corresponding mechanisms of release, where hyperacetylated HMGB1 undergoes regulated secretion from immune cells during inflammation [3], and hypoacetylated HMGB1 is passively released from necrotic cells after tissue injury [35]. Similarly, secretion of the HMGB1 proteoform with an intramolecular disulfide between Cys-23 and Cys-45 (Figure 1) has been associated with inflammation and oxidative stress, whereas reduced HMGB1 with no intramolecular disulfide bond has been associated with passive release from necrotic cells [36,37]. However, many of the studies supporting this conceptual framework have been discredited or retracted [reviewed in [28]]. Therefore, we considered that it was important to definitively characterize distinct patterns of HMGB1 PTMs arising from chemotherapy-induced HMGB1 secretion.

PTMs could play a role in mediating the mobility of HMGB1 in cisplatin-treated cells. For example, HMGB1 is a substrate of poly [ADP-ribose] polymerase 1 (PARP1) [38,39], and DNA damage is a potent inducer of PARP1 activity [38]. Therefore, secreted HMGB1 could conceivably be ADP-ribosylated prior to secretion. Similarly, in vitro experiments have demonstrated that acetylation and phosphorylation of HMGB1 can modulate its DNA-binding properties [40,41]. Furthermore, the binding affinity between cisplatin-modified DNA and reduced HMGB1 is 10-fold greater than with oxidized HMGB1 in vitro [42], which suggests that differences in cellular redox state could contribute to variability in cellular response to platinum drugs. Therefore, characterizing the PTMs and oxidation state of HMGB1 is necessary to fully understand the mechanism by which it is secreted from NSCLC cells resulting from platinum drug treatment.

In the current study, we determined whether platinum drugs could induce regulated HMGB1 translocation and secretion in the human A549 NSCLC cell line. We report a sensitive and specific assay to quantify HMGB1 proteoforms in cells and cell culture media using immunoprecipitation (IP) with magnetic beads covalently crosslinked with dimethyl pimelimidate (DMP) to an HMGB1 polyclonal antibody (pAb) coupled with nano liquid chromatography parallel reaction monitoring/high-resolution mass spectrometry (LC-PRM/HRMS). A differential alkylation approach with iodoacetamide (IAA) and N-ethylmaleimide (NEM) modification of Cys residues was used in combination with a [^13^C^15^N]-HMGB1 internal standard to characterize and accurately quantify the redox state of HMGB1 Cys residues. Post-IP acetylation with *D*6-acetic anhydride was used to convert all acetylated and nonacetylated HMGB1 into a single molecular form, to characterize and accurately quantify endogenous acetylation of HMGB1 lysine residues. These sensitive and specific methods revealed, surprisingly, that cisplatin, but not carboplatin or transplatin, could induce HMGB1 release from A549 cells prior to cell death or loss of plasma membrane integrity. Furthermore, the secretion was mediated by nuclear transport rather than by acetylation, ADP-ribosylation, phosphorylation, or oxidation.

## 2. Materials and Methods

### 2.1. Chemicals and Reagents

Reagents and solvents were LC/MS-grade quality unless otherwise noted. Burdick and Jackson (Muskegon, MI, USA) supplied LC/MS grade water and acetonitrile. Cambridge Isotope Laboratories (Andover, MA, USA): [^13^C_6_^15^N_2_] lysine, [^13^C_9_^15^N_1_] tyrosine and [^2^H_6_] acetic anhydride. EMSCO/FISHER (Philadelphia, PA, USA) supplied Pierce Protein A/G magnetic agarose beads, formic acid, Dulbecco’s phosphate buffered saline (PBS). Fisher Scientific (Pittsburgh, PA, USA) supplied ammonium hydroxide (Optima). MilliporeSigma (Billerica, MA, USA) supplied the anti-HMGB1 rabbit pAb raised against the C-terminal acidic tail of HMGB1 (H9539), ethylenediaminetetraacetic acid (EDTA)-free protease inhibitor cocktail, L-cysteine, chloramine-T hydrate, triethanolamine, and ethanolamine. Abcam (Waltham, MA, USA) supplied the rabbit anti-HMGB2 monoclonal antibody (mAb) raised against an N-terminal HMGB2 peptide of unspecified sequence (ab124670). Anti-histone H3 rabbit pAb (cat. # PA5-16183) and anti-fatty acid synthase (FASN) rabbit pAb (cat. # MA5-14887) were purchased from Invitrogen (Carlsbad, CA, USA). Novus Biologicals (Centennial, CO, USA) supplied early apoptosis-specific stain polarity Sensitive Indicator of Viability and Apoptosis (pSIVA) and propidium iodide (PI) fluorescent stains. Promega (Madison, WI, USA) supplied mass spectrometry grade trypsin endo-proteinase. Thermo Fisher Scientific (Waltham, MA, USA) supplied tris(2-carboxyethyl)phosphine (TCEP), endoproteinase Glu-C from *Staphylococcus aureus* V8, Tris HCl pH 8 solution, and dimethyl pimelimidate (DMP). Horseradish peroxidase (HRP) was supplied by Santa Cruz Biotechnology, (Dallas, TX, USA). Perkin Elmer (Waltham, MA, USA) supplied the Western Lightning Plus electrochemical luminescence (ECL) reagent. REN and EMMESO malignant mesothelioma cells were a kind gift from Dr. Steven M. Albelda, University of Pennsylvania. Platinum drugs were dissolved in PBS because dimethyl sulfoxide (DMSO) can cause oxidative stress and apoptosis [43,44,45].

### 2.2. Platinum Treatments and Lung Cancer Cell Models

A549 cells (American Type Culture Collection) were cultured in Dulbecco’s Modified Eagle’s Medium/Ham’s F-12 Nutrient Mixture media with 15 mM HEPES and 2 mM glutamine (Thermo Fisher Gibco, 11330-032), supplemented with 1% Penicillin-Streptomycin (Thermo Fisher Gibco 15140-122) and 5% FBS (Sigma Aldrich F2442). Cells were seeded at a density of 3 × 10^6^ cells per 35 × 10 mm plate and cultured at 37 °C with 5% CO_2_. Cisplatin, transplatin, and carboplatin were prepared as 5 mM stocks in PBS with 30 s pulse sonication and diluted into 10 mL cell culture media for treatment. After 24 h, media was collected and subjected to centrifuge to remove any cell debris (1000 g, 10 min, 4 °C) and the supernatant was transferred to fresh 15-mL tubes. REN and EMMSO cell lines were cultured in RPMI Media with 2 mM glutamine, supplemented with 1% Penicillin-Streptomycin and 5% FBS.

### 2.3. Cell Culture Media Western Blots and Fluorescent Cell Staining

After collection, aliquots (100 µL) of A549 media were dried under N_2_ gas and prepared in LDS sample buffer for electrophoresis separation and membrane transfer. HMGB1 was detected via an HMGB1 rabbit pAb (ab79823) and antirabbit HRP secondary antibody and visualized with an ImageQuant LAS 4000 camera. HMGB2 was similarly detected via an HMGB2 rabbit pAb (ab124670) and antirabbit HRP. Western blots were developed with ECL reagents. Fluorescent staining for nuclear DNA, necrosis, and apoptosis was done with Hoechst 33342, pSIVA, and PI dyes (Novus, NBP2-29382), respectively, following washing of live cells with PBS. Fluorescent microscopy was performed with BioTek Gen5 3.04 Imager. Images were generated from overlaying the blue, green, and magenta filters, respectively.

### 2.4. Expression and Purification of Stable Isotope Labeling by Amino Acids in Cell Culture (SILAC) Labeled HMGB1

HEK293T cells were cultured in DMEM/F12 SILAC medium containing 0.5 mM [^13^C_6_^15^N_2_]-lysine and 0.2 mM [^13^C_9_^15^N_1_]-tyrosine for at least three passages. HMGB1-FLAG tag plasmid (Origene RC205918) was transfected into HEK293T cells by use of lipofectamine 3000 transfection reagent (Invitrogen) following the manufacturer’s instructions. Transfected cells were harvested in NP-40 lysis buffer [150 mM NaCl, 50 mM Tris-HCl, pH 7.5, 1 mM EDTA, 0.5% Triton X-100, 0.5% NP-40, and 1 mM DTT, dithiothreitol (DTT)] containing protease inhibitor cocktail after 48 h transfection. Harvested cells were subjected to pulse sonication on ice (30 s) and centrifuged (17,720 g, 15 min, 4 °C) before cell lysate supernatant was carefully transferred to fresh tubes. Anti-FLAG M2 magnetic beads (Sigma Aldrich) were washed twice with NP-40 lysis buffer and added to each tube and incubated overnight at 4 °C with gentle rotation. After removal of the supernatant, the beads were washed with the following solutions sequentially: NP-40 lysis buffer × 3, 0.5 M KCl in NP-40 lysis buffer × 1, 1 M KCl in NP-40 lysis buffer × 1, and PBS × 2. SILAC HMGB1 was eluted from beads with glycine elution buffer (0.1 M glycine-HCl, pH 2.5) for 20 min and subsequently with 10 mM NH_4_HCO_3_ for 10 min. Aliquots of 250 mM NH_4_HCO_3_ were added to pooled eluent until pH was neutral.

### 2.5. Crosslinked Magnetic Bead Immunoprecipitation and Trypsin Digestion

Protein A/G magnetic beads were gently vortexed and 40 μL dispersed beads were transferred to two 2-mL protein LoBind microcentrifuge tubes (Eppendorf, 022431102). Beads were washed twice with PBS and twice with Buffer A (0.1 M sodium phosphate, pH 7.4). Each tube was incubated with 100 μL buffer A and 100 μL of either H9539 (Sigma Aldrich) or ab79823 (Abcam) anti-HMGB1 rabbit pAb for 2 h at 4 °C with gentle rotation, followed by the addition of 400 μL buffer A and further overnight incubation at 4 °C. The pAb solutions were removed and beads were gently washed twice with buffer A and then twice with 1 mL of cross-linking buffer (0.2 M triethanolamine, pH 8). After washes, the beads were suspended in 1 mL of 25 mM DMP prepared in cross-linking buffer and incubated at room temperature for 1 h with gentle rotation. The DMP solution was removed, and the beads were washed with 1 mL of blocking buffer (0.1 M ethanolamine, pH 8.2) and then incubated at room temperature for 30 min in 1 mL of blocking buffer. The blocking buffer was then removed, and the beads were washed twice with PBS. The PBS was removed, and the beads were incubated in glycine elution buffer for 15 min at room temperature with gentle rotation to remove any residual non-crosslinked antibody from the beads. After removing the glycine elution buffer and two final washes with PBS, beads with both HMGB1 pAbs were equally aliquoted into fresh protein LoBind tubes containing 1 mL A549 media, SILAC FLAG-HMGB1 internal standard (800 ng) and 100 μL IP lysis buffer. Media immunoprecipitation samples were incubated overnight at 4 °C with gentle rotation and gently washed twice with PBS. HMGB1 was eluted with 15 min vigorous shaking of beads in 100 μL elution buffer and then with 5 min vigorous shaking 50 μL 10 mM NH_4_HCO_3_. The two eluents of each sample were combined and subsequently neutralized with 50 μL 250 mM NH_4_HCO_3_. The combined elute was dried under N_2_ for subsequent resuspension in 25 mM NH_4_HCO_3_ pH 8 and digested with 400 ng trypsin protease per sample overnight at 37 °C with 600 RPM shaking. We have also previously shown that the IP procedure can efficiently isolate acetylated forms of HMGB1 [15].

### 2.6. Cell Fractionation

Nuclear and cytosolic subcellular fractions were prepared with ProteoExtract Subcellular Proteome Extraction Kit (Millipore Sigma, 539790) following a protocol adapted from the manufacturer’s instructions. After removing cell culture media, cell plates were gently washed twice with PBS and aspiration. Cells were gently lifted in 1 mL PBS and pelleted by centrifugation (600× *g*, 6 min, 4 °C). The PBS supernatant was discarded, and cells were resuspended in 300 µL wash buffer and gently rocked (5 min, 4 °C) before centrifugation (600× *g*, 6 min, 4 °C). This entire wash step was repeated, and then the cells were resuspended in 200 µL extraction buffer I with 2 µL protease inhibitor and gently rocked (5 min, 4 °C) to extract the cytosolic fraction. The cell sample was subjected to centrifugation (1000× *g*, 10 min, 4 °C) and the supernatant was transferred to a clean tube to provide the cytosolic fraction. The remaining pellet was resuspended in 200 µL provided Extraction Buffer II with 2 µL provided protease inhibitor and gently rocked (30 min, 4 °C) before centrifugation (6000× *g*, 10 min, 4 °C). The supernatant was discarded, and the remaining pellet was resuspended in 100 µL extraction buffer III with 2 µL d protease inhibitor and 1 µL benzonase nuclease. The cell sample was gently rocked (10 min, 4 °C) to extract the nuclear fraction. Finally, the sample was subjected to centrifugation (7000× *g*, 10 min, 4 °C) and the supernatant was transferred to a clean tube to provide the nuclear fraction. Cytosolic and nuclear fraction were tested with immunoblotting for FASN and Histone-3/Histone-4 antibodies, respectively.

### 2.7. In-Solution D6-Acetic Anhydride D_3_-Acetylation and GluC and Chymotrypsin Digestion

Following glycine-HCl and NH_4_HCO_3_ elution of HMGB1, 100 μL 250 mM NH_4_HCO_3_ was added to each sample to elevate pH to 8.5. A working stock of *D*6-acetic anhydride (Cambridge Isotopes DLM-1162-PK) was diluted 14X in acetonitrile. Successive additions of 30 µL ammonium hydroxide (Fisher A470) and 140 µL working stock *D*6-acetic anhydride were added to each sample and incubated for 1 h at 37 °C with 600 RPM shaking. Following the acetylation reaction, samples were dried under N_2_ gas and resuspended in 200 μL 50 mM NH_4_HCO_3_ pH 8 for Glu-C protease digestion or 200 μL pH 8 for Chymotrypsin digestion. 

### 2.8. IAA/NEM Cys Derivatization

SILAC FLAG-HMGB1 stock was reduced in 50 mM TCEP (Thermo Scientific 77720) for 1 h at pH 7 and 30 °C with 600 RPM shaking. Then stock was diluted to 20 ng/µL with Optima water. Iodoacetamide was prepared away from light as a 500 mM stock in 250 mM NH_4_HCO_3_. Reduced SILAC FLAG-HMGB1 internal standard (400 ng) was added to each 1-mL A549 media sample, followed by addition of iodoacetamide solution for a 60 min reaction with 50 mM iodoacetamide at pH 8.5 and 25 °C in the dark with 600 RPM shaking. Residual iodoacetamide was then quenched with 100 mM L-Cys at pH 8.5 and 30 °C in the dark with 600 RPM shaking and the sample pH was brought to neutral with 0.5 M HCl (Fisher SA48-500). Then more reduced SILAC internal standard (400 ng) was added to each sample, followed by 100 µL IP lysis buffer for 1 mM BME final concentration. Both HMGB1 pAbs were equally aliquoted into samples for overnight immunoprecipitation and subsequent elution as previously described [15]. Following elution, samples were reduced with a large excess (25 mM) TCEP for 1 h at pH 6 and 30 °C with 600 RPM shaking. The high concentration of TCEP prevented its consumption by any NAD^+^ that might have been present in the eluate [46]. N-ethylmaleimide was prepared in the dark as a 300 mM stock in water, and aliquots were added to each sample for a 60 min reaction with 30 mM N-ethylmaleimide at pH 7 and 25 °C in the dark with 600 RPM shaking. Samples were then dried under N_2_ gas and tryptic digests were prepared as previously described [15]. Glu-C digests were prepared following *D*6-acetic anhydride acetylation steps as previously described [15].

### 2.9. NanoLC-PRM/HRMS

UHPLC-MS was conducted on a Q Exactive HF hybrid quadrupole-Orbitrap mass spectrometer coupled to a Dionex Ultimate 3000 RSLCnano with capillary flowmeter chromatographic system (Thermo Fisher Scientific, San Jose, CA, USA). The nanoLC system was composed of two columns, including a trapping column (Acclaim PepMap C18 cartridge (0.3 mm × 5 mm, 100 Å, Thermo Scientific) for preconcentration purposes and an analytical column (C18 AQ nano-LC column with a 10 μm pulled tip (75 μm × 25 cm, 3 μm particle size; Columntip, New Haven, CT, USA) to separate digested peptides; two pumps, including one nanopump delivering solvents to analytical column and a micropump connecting to the trapping column; and a 10-port valve. The nanoLC system was controlled by Xcalibur software from the Q-Exactive mass spectrometer. Samples (8 μL) were injected via microliter-pickup injection mode. Loading solvent was water/acetonitrile (99.7:0.3 *v*/*v*) containing 0.2% formic acid. A 10-port valve was set at the loading position (1–2) with the loading solvent at 10 μL/min for 3 min. The valve was then changed to the analysis position (1–10), at which time the trapping column was connected with the analytical column, and samples loaded on the loading column were back-flushed into the analytical column. The valve was maintained in the analysis position for 10 min before the end of the run, when it was switched to the loading position ready for the next analysis. Samples were eluted with a linear gradient at a flow rate of 0.35 μL/min: 2% B at 2 min, 5% B at 15 min, 35% B at 40 min, 95% B at 45–55 min, 2% B at 58–70 min. Solvent A was water/acetonitrile (99.5:0.5 *v*/*v*) containing 0.1% formic acid, and solvent B was acetonitrile/water (98:2 *v*/*v*) containing 0.1% formic acid. Nanospray Flex ion source (Thermo Scientific) was used. MS operating conditions were as follows: spray voltage 2500 V, ion transfer capillary temperature 250 °C, ion polarity positive, S-lens RF level 55, in-source collision-induced dissociation (CID) 1.0 eV. Both full-scan and parallel reaction monitoring (PRM) were used. The full-scan parameters were resolution 60,000, automatic gain control (AGC) target 1 × 10^6^, maximum IT 200 ms, scan range *m*/*z* 290–1600. The PRM parameters were resolution 60,000, AGC target 2 × 10^5^, maximum IT 80 ms, loop count 5, isolation window 1.0 Da, NCE 25.

### 2.10. Quantification of HMGB1

Standard curve samples for tryptic and Glu-C digests of HMGB1 were prepared in media in triplicate. Recombinant HMGB1 was serially diluted in water and added to each media sample, followed by 800 ng SILAC internal standard. Two peptides per digest with three of the best nano-LC-PRM/HRMS transitions (H^31^PDASVNFSEFSK^43^ and G^115^EHPGLSIGDVAK^127^) were used to generate the standard curves with mean values of light-to-heavy ratios of peak areas of the non-Cys tryptic peptides (Appendix A) from cisplatin-treated A549 media. A549 media HMGB1 was quantified by light-to-heavy ratios of peak areas using the generated standard curves (Appendix A).

### 2.11. Quantification of HMGB1 Lysine Acetylation and Cys Oxidation

Lysine acetylation was quantified as previously described [15]. Briefly, PRM transitions for all possible CH_3_-acetylated and *CD*_3_-acetylated lysine residues were selected in the relevant samples from Glu-C and Chymotrypsin digests. The ratio of each light peptide peak area to the corresponding SILAC peptide peak area was calculated by Xcalibur QuanBrowser and qualified by Skyline. Cys oxidation was quantified by comparing the light-to-heavy ratio of peak areas in CAM- and NEM-modified Cys-containing peptides in the relevant trypsin and Glu-C digests. CAM- and NEM-modified Cys-containing peptides were confirmed by MS2 mass spectra (Appendix A). The CAM ratio compared to NEM ratio for a given Cys-containing peptide represents the ratio of reduced to disulfide oxidized Cys. The percentage of disulfide oxidation of a given Cys residue was calculated by NEM L/H divided by the sum of CAM L/H and NEM L/H and averaged for three PRM transitions per peptide.

### 2.12. Inhibitor Co-Treatments and ImageJ Blot Quantification

A549 cells were seeded in 6-well cell culture plates at 0.5 × 10^6^ cells per well. Olaparib, GO-6983, Rottlerin, and KPT-330 were prepared as stocks in DMSO and LMB was provided in a 70/30 Methanol/Water (*v*/*v*) solution. Cell culture media was replaced fresh in each well right before inhibitor solutions and vehicle controls were added to each well. Following 3 h in the cell culture incubator, cisplatin or PBS vehicle control was added to each well for 24 h incubation. After 24 h, media was collected and subjected to centrifuge to remove any cell debris (1000 g, 10 min, 4 °C) and the supernatant was transferred to fresh 2-mL tubes. Western blots for HMGB1 were performed as previously described [15]. Following imaging, Western blots were quantified with ImageJ software by subtracting the membrane background signal from each blot using the same area and normalizing the signal to the calculated signal from the blot of HMGB1 (50 ng) on the membrane. 

### 2.13. Trypan Blue Exclusion Assay for A549 Proliferation and Viability

Trypan blue exclusion assays were performed on A549 cells evenly split into 35 × 10 mm plates for a target density of 3 × 10^6^ cells per plate. After all cells had re-adhered, they were treated with either cisplatin or PBS vehicle control (N = 5). For the trypan blue exclusion assays, cell culture media was aspirated, and cells were lifted by gentle washing with 2.5 mL trypsin-EDTA 0.5% and 2.5 mL PBS. Aliquots of 200 µL cell suspension were mixed with 200 µL trypan blue solution, and 2 × 11 µL of the mixture was then added to cell counter slides for analysis for cell count and viability by an automated cell counter (LUNA Automated Cell Counter L10001, Logos). The values obtained from the A and B sides of each slide were averaged and used to calculate the total viable and dead cells on a given plate. 

### 2.14. Asp-N Protease Digests of A549 Subcellular Fractions

A549 subcellular fractions were collected as previously described [47]. Partial aliquots were taken from each fraction for sodium dodecyl sulfate polyacrylamide gel electrophoresis (SDS-PAGE) sample preparation. SILAC HMGB1 was distributed equally to each partial aliquot. SDS-PAGE was run as previously described [15], followed by overnight Colloidal Blue staining (Thermo Fisher, LC6025). Following staining, gels were briefly rinsed with deionized water, and gel pieces between 25 kDa and 37 kDa were excised from each lane to cut into approximately 1 mm^3^ pieces and added to fresh microcentrifuge tubes. Gel pieces were rinsed by gentle shaking in 25 mM ammonium bicarbonate in 50% acetonitrile by volume and the solution aspirated before the pieces were dried with 200 µL aliquots of acetonitrile and gentle shaking. After drying the acetonitrile was aspirated and 5 µL *D*6-acetic anhydride was added to each tube of gel pieces to absorb, followed by 50 µL of 100 mM ammonium bicarbonate. After absorption, 150 µL of 100 mM ammonium bicarbonate and 15 µL were added to each tube to bring the pH to approximately 8.5. Samples were incubated for 1 h at 37 °C with 600 RPM shaking, followed by removal of the supernatant, rinsing again with 25 mM ammonium bicarbonate in 50% acetonitrile, and drying again with acetonitrile. A working stock of Asp-N protease was prepared in water 25 ng/µL. After removing acetonitrile, 30 µL (750 ng) Asp-N protease was added to each tube for absorption by the gel pieces. Following absorption, 200 µL 25 mM ammonium bicarbonate was added to each tube and samples were incubated overnight at 37 °C with 600 RPM shaking. Following incubation, the digest supernatant was removed and added to fresh tubes. Residual peptides were extracted from the gel pieces by 45 min bath sonication with 150 µL extraction buffer (3% formic acid in 50% aqueous acetonitrile). The supernatants from this extraction were recombined with the corresponding digest supernatants, and the samples were dried under N_2_ gas and prepared for LC-MS analysis as previously described [15].

## 3. Results

### 3.1. Cisplatin Induces Regulated HMGB1 Secretion from Lung Cancer Cells

By examining anti-HMGB1 immunoblots of cell culture media, we found that HMGB1 was secreted by the A549 NSCLC cell line with cisplatin treatment in a dose-dependent relationship (Figure 2A). Among the tested platinum chemotherapeutics, cisplatin was the most effective, with the stereoisomer transplatin and the analogue carboplatin showing little effect on HMGB1 release. Conversely, DMSO, an inducer of oxidative stress [43] and apoptosis [44,45], also induced extracellular release of HMGB1. Mesothelioma cell lines REN and EMMESO both constitutively secreted HMGB1 in the absence of platinum drugs, but A549 cells showed no such constitutive secretion (Figure 2A). To accurately quantify cisplatin-secreted HMGB1 from the A549 NSCLC cells, we used a nanoLC-PRM/HRMS platform to analyze tryptic HMGB1 peptides [15].

SILAC methodology was applied to generate lysine- and tyrosine-labeled SILAC (K*Y*-SILAC) HMGB1, which was then used as the internal standard for quantification of HMGB1 isolated from cell culture media by IP, then subjected to trypsin digestion and nanoLC-PRM/HRMS analysis. Calibration curves were made with recombinant HMGB1 and SILAC HMGB1 for tryptic peptide nanoLC-PRM/HRMS transitions H^31^PDASVNFSEFSK^43^ dehydration, H^31^PDASVNFSEFSK^43^ y12, G^115^EHPGLSIGDVAK^127^ b7, and G^115^EHPGLSIGDVAK^127^ y10 (Appendix A). NanoLC-PRM/MS analysis confirmed the immunoblot data by showing that there was dose-dependent secretion of HMGB1 with cisplatin at 20 μM [551.4 ± 76.3 ng; mean ± standard deviation (SD)] and 100 μM (1139 ± 102 ng) (Figure 2B). This was significantly different from PBS (204.9 ± 3.3 ng) at 20 μM (*p* = 0.0014) and 100 μM (*p* < 00001), respectively (Figure 2B). There was also increased HMGB1 secretion with DMSO (345.8 ± 30.5, *p* < 0.0014) when compared with PBS, and there was a minimal effect with other cisplatin analogs that were tested (Figure 2B).

### 3.2. Specific Cisplatin-Mediated Secretion of HMGB1 from Lung Cancer Cells

To evaluate if cisplatin treatment specifically induced the extracellular release of HMGB1, and not all nuclear proteins, we examined the nuclear, cytosolic, and media fractions of A549 cells for HMGB1 and HMGB2. Anti-HMGB1 and anti-HMGB2 immunoblotting showed specificity for HMGB1 (Appendix A) and HMGB2 (Appendix A), respectively, as demonstrated with recombinant proteins (Appendix A). Although HMGB1 and HMGB2 were both abundant nuclear proteins, there was much more HMGB1 in A549 cytosol (Appendix A) than HMGB2 (Appendix A). Cisplatin treatment induced significant secretion of HMGB1 into the A549 cell culture media (Appendix A), but no secretion of HMGB2 (Appendix A). We also assessed the subcellular fractions by anti-histone H4 and anti-FASN immunoblotting (Appendix A, which confirmed robust enrichment of the nuclear and cytosolic fractions and provided evidence that there was no evidence of leakage of histone H4 or FASN into the cell culture media with cisplatin treatment (Appendix A).

To provide orthogonal analysis of HMGB1 and HMGB2 subcellular distribution with cisplatin treatment, we performed qualitative nano-LC-HRMS/MS analysis of HMGB1- and HMGB2-specific Asp-N peptides (Appendix A). Nuclear, cytosolic, and extracellular media fractions from cisplatin-treated A549 cells were spiked with SILAC HMGB1, separated, and prepared for Asp-N protease digestion. Asp-N was carefully chosen to generate multiple peptides that could be used to distinguish HMGB1 from HMGB2 in the subcellular fraction samples, and *CD*_3_-acetylation (Ac*) was used to improve the chromatography of these peptides. We compared HMGB2 Asp-N peptides D^124^TAKKLGEMWSEQSAK^139^, D^140^KQPYEQKAAKLK^152^, and D^158I^AAYRAKGKS^168^ with HMGB1 Asp-N peptides D^124^VAKKLGEMWNNTAA^138^, D^139^DKQPYEKKAAKLK^152^, and D^158^IAAYRAKGKP^168^, respectively, in the nuclear (Appendix A) and cytosolic fractions (Appendix A) as well as the media (Appendix A). Although differences between HMGB1 and HMGB2 sequences precluded a quantitative comparison in this context, the results confirmed that abundant HMGB1 and HMGB2 peptides were present in the nuclear fraction after protease digestion (Appendix A). However, only HMGB1 peptides were present after protease digestion of the A549 cytosol (Appendix A) or the media after cisplatin treatment of A49 cells (Appendix A). No signals for HMGB2 peptides were detected in the A549 cytosol or A549 cell media after cisplatin treatment (Appendix A, blue arrows).

### 3.3. HMGB1 Secreted from Cisplatin-Treated Lung Cancer Cells Is Not Acetylated

To investigate the mechanisms regulating cisplatin-mediated HMGB1 secretion, we first decided to probe acetylation of lysine residues on secreted HMGB1. In order to quantify site-specific acetylation, we adapted a previously reported strategy using HMGB1 IP and nanoLC-PRM/HRMS analysis where we showed that acetylated HMGB1 is isolated by the IP procedure [15]. SILAC HMGB1 internal standard was added to cell culture media samples collected from A549 cells, and light and heavy HMGB1 was immunopurified from these samples (Figure 1). These samples were treated with *D*6-acetic anhydride to *CD*_3_-acetylate all unmodified HMGB1 lysine residues. This process converts all HMGB1 proteoforms to one universally acetylated molecular form, only differing in mass of the acetyl-lysine modifications. We primarily targeted the most often reported regions of HMGB1 acetylation in other models, including NLS1 (K28-30), NLS2 (K180-185) and the N-terminal region (K3-K12), and we designed our PRM/MS method to analyze all possible combinations of acetylation within them.

NanoLC-PRM/HRMS analysis of the acetylation state of NLS1, NLS2, and N-terminal region of secreted HMGB1 is shown in the chromatograms in Figure 3A–C. Chromatograms for the b11 transitions from the NLS1 Glu-C peptide HK^28^K^29^K^30^HPDASVNFSE revealed no signal for triple, double, or single CH_3_-acetylation of this peptide (Figure 3A). In contrast, signals were observed from the fully *CD*_3_-acetylated peptide (Figure 3A), which confirmed that K-28, K-29, and K-30 all had no endogenous acetylation. Similarly, chromatograms of the b4 transitions from the NLS2 Glu-C peptide K^180^SK^182^K^183^K^184^K^185^E revealed no CH_3_-acetylation for any of the peptide’s five lysine residues, with multiple transitions analyzed for the same parent ion to check for possible CH_3_-acetylation on either MS fragment produced by the b4 transition of this peptide (Figure 3B). As with the NLS1 peptide, K^180^SK^182^K^183^K^184^K^185^E was fully *CD*_3_-acetylated, confirming that HMGB1 K180, K182, K183, K184, and K185 all had no endogenous CH_3_-acetylation. Additional MRM transitions were used to confirm universal *CD*_3_-acetylation of NLS1 (b10, b11, b12; Appendix A) and NLS2 (b3, b4, b5; Appendix A), as shown in the relevant chromatograms. Chromatograms of the y8 transitions for the N-terminal chymotrypsin peptide GK^3^GDPK^7^K^8^PRGK^12^MSSY (Figure 3C) also showed no endogenous CH_3_-acetylation for any of its four lysine residues or the N-terminal amino moiety. We again used multiple transitions analyzed for the same parent *m*/*z* to check for possible CH_3_-acetylation on either MS fragment produced by the y8 transition of this peptide (Figure 3C). As with the NLS peptides, GK^3^GDPK^7^K^8^PRGK^12^MSSY was fully *CD*_3_-acetylated, indicating that HMGB1 K3, K7, K8, K12, and the N-terminal amino moiety of HMGB1 all had no endogenous acetylation with cisplatin-mediated secretion (Figure 3C).

### 3.4. Non-Targeted Analysis of Acetylation Sites on HMGB1 Secreted from Cisplatin-Treated Lung Cancer Cells

In addition to the targeted nanoLC-PRM/HRMS analysis of acetylation in the specific regions described above, we wanted to apply full-scan nanoLC-high resolution tandem mass spectrometry (HRMS/MS) analysis to other lysine residues within secreted HMGB1. We carefully chose Glu-C, chymotrypsin, and Asp-N proteases to generate as many possible lysine-containing peptides to with maximal coverage of HMGB1′s 43 lysine residues (Figure 1). In total, we were able to probe acetylation on 34 of HMGB1′s 43 lysine residues by nanoLC-HRMS/MS, the results of which are summarized in Appendix A. We confirmed each peptide with the corresponding heavy peptide generated from SILAC HMGB1, and we checked for possible CH_3_-acetylation by analyzing all possible product ions. Therefore, we found no evidence of endogenous acetylation at any lysine residue in HMGB1 secreted with cisplatin treatment. Furthermore, HMGB1 secreted with DMSO treatment also showed no sign of endogenous acetylation (Appendix A). The data were essentially the same as those discussed above for cisplatin-mediated secretion of HMGB1 (Figure 3A–C).

### 3.5. Cisplatin Mediates Secretion of Reduced HMGB1

We next interrogated the redox status of the Cys residues on HMGB1 in A549 cell culture media. To distinguish between free (reduced) and disulfide (oxidized) Cys residues on HMGB1, we used a differential alkylation approach with iodoacetamide (IAA) and N-ethylmaleimide (NEM), in which Cys disulfides that were precluded from forming carbamidomethyl (CAM) derivatives with IAA were reduced to free thiols with TCEP prior to the NEM reaction (Figure 2). A549 media samples were subjected to HMGB1 IP and elution for downstream protease digestion and nanoLC-PRM/HRMS analysis. To control for variation in reaction completeness, IP recovery, efficiency of protease digest, and ionization efficiency of modified peptides among all samples, we used two aliquots of SILAC HMGB1 internal standard, added separately for each of the IAA and NEM reaction steps. Following IAA reaction, residual IAA was quenched to prevent the second aliquot of SILAC HMGB1 internal standard from reacting with IAA on Cys residues. A second aliquot of SILAC HMGB1 was added prior to HMGB1 immunopurification to ensure that ratios of CAM-modified and NEM-modified HMGB1 peptides were not affected by variation in IP and elution efficiency among samples or batches. Reactions with TCEP and NEM were performed post-elution following IP to remove IAA and Cys from the samples. Although trypsin digestion generated sufficient peptides for analysis of Cys-23 with MSSYAFFQTC^23^R and Cys-106 RPPSAFFLFC^106^SEYRPK, no tryptic peptide was obtained for analysis of Cys-45 oxidation. 

We therefore prepared a second set of A549 media samples in parallel, in which we performed in-solution *CD*_3_-acetylation with *D*6-acetic anhydride and carefully chose Glu-C protease digestion to generate enough peptide for analysis of the Cys-45 in FSKKC^45^SE. *CD*_3_-acetylation of the lysine residues was used to significantly improve chromatography as well as to facilitate the concurrent analysis of lysine acetylation on this peptide. We found that, in comparison to other treatment conditions, cisplatin induced the secretion of significantly more fully reduced HMGB1, as defined by the redox status of Cys-23, Cys-45, and Cys-106 (Figure 1). There was 228 ± 39.6 ng and 353 ± 44.4 ng of reduced HMGB1 secreted at 20 μM and 100 μM cisplatin, respectively (Figure 4A). This result was significantly different from PBS (59.8 ± 3.1 ng) at both 20 μM (cisplatin *p* = 0.0039) and 100 μM cisplatin (*p* = 0.0007). Although DMSO treatment also increased HMGB1 secretion compared with PBS (Figure 2B), the amount of reduced HMGB1 (37.9 ± 3.1) was slightly lower than that secreted by PBS (Figure 4A). In contrast, there was significantly more oxidation at Cys-23 (*p* < 0.0008, *p* < 0.0026; Figure 4(Ba)), Cys-45 (*p* < 0.001, *p* < 0.001; Figure 4(Bb)), and Cys-106 (*p* < 0.0014, *p* < 0.0006; Figure 4(Bc)) after DMSO treatment than the HMGB1 secreted by 20 μM and 100 μM cisplatin, respectively.

Across all treatments, Cys-23 and Cys-45 were oxidized to a similar extent (Figure 4(Ba,Bb)) but there was also more oxidization at these residues than at Cys-106 (Figure 4(Bc)). NanoLC-PRM/HRMS analysis of the redox status of Cys-containing peptides of secreted HMGB1 is shown in the chromatograms in Figure 5. For HMGB1 secreted with cisplatin treatment, assigned MS/MS spectra for Cys-23, Cys-45, and Cys-106 peptides are shown in Appendix A, for CAM and NEM modifications, respectively. Chromatograms in Figure 5A show the y6, y7, y8, and b2 transitions of tryptic peptide MSSYAFFQTC^23^R with CAM (left panel) and NEM (right panel) modifications. The relevant ratios of the area under the curves (AUCs) reflect the amount of reduced Cys-23 (CAM) peptide compared with the oxidized Cys-23 (NEM) peptide (Figure 5A).

Chromatograms of the heavy MSSYAFFQTC^23^R transitions produced from SILAC HMGB1 were used to calculate light-to-heavy ratios of the AUCs for each corresponding CAM- and NEM-modified peptide transition (Figure 5A). The mean of four MRM transitions (b2, y6, y7, y8) in Figure 5A was used to calculate the oxidized Cys-23 (disulfide) percentage shown in Figure 4B. Similarly, chromatograms show the b4, y4, and b3 MRM transitions of Glu-C peptide FSKKC^45^SE with CAM (left panel) and NEM (right panel) modifications. The relevant AUC ratios reflect the amount of reduced Cys-45 (CAM) peptide compared with the oxidized Cys-45 (NEM) peptide (Figure 5B). The mean of all three MRM transitions was used to calculate the Cys-45 disulfide oxidation percentage in Figure 4B. Chromatograms also show the y7, y6, b7, and b6 transitions of tryptic peptide RPPSAFFLFC^106^SEYRPK with CAM (left panel) and NEM (right panel) modifications. The relevant AUC ratios reflect the amount of reduced Cys-106 (CAM) peptide compared with the oxidized Cys-106 (NEM) peptide (Figure 5C). A mean of the three MRM transitions (b3, b5, y4) shown in Figure 5C was used to calculate the oxidized Cys-106 (disulfide) peptide percentage shown in Figure 4B.

### 3.6. Nuclear Exportin 1 (XPO1) Regulates Cisplatin-Mediated HMGB1 Secretion

We investigated whether cisplatin-mediated HMGB1 secretion is regulated by other post-translational modifications by examining the effects of different pharmacological inhibitors on this process. A549 cells were pre-treated with a different inhibitor or vehicle control, followed by treatment for 24 h with either 50 µM cisplatin or PBS vehicle control, and the cell culture media was assayed by anti-HMGB1 immunoblot (Figure 6). We found that PARP1 inhibitor Olaparib (Figure 6A) had no effect on cisplatin-mediated HMGB1 secretion. Similarly, pan-PKC inhibitors GO-6983 (Figure 6B) and Rottlerin (Figure 6C) had no effect on HMGB1 secretion. Conversely, XPO1 (chromosomal maintenance 1; CRM1) inhibitors leptomycin B (LMB, Figure 6D) and KPT-330 (Figure 6E) significantly attenuated cisplatin-induced HMGB1 secretion from A549 cells. Relative quantification of these immunoblots showed that LMB and KPT-330 inhibited cisplatin-mediated HMGB1 secretion by 95% (Figure 6F) and 52%, (Figure 6G), respectively (*p* = 0.0012 and *p* = 0.0030, respectively; Figure 6F,G).

### 3.7. Inhibition of Nuclear Exportin 1 (XPO1) Increased Cisplatin-Mediated Nuclear HMGB1 Accumulation and Decreased HMGB1 Secretion

Western blot analysis had revealed that HMGB1 was present in the nucleus and cytosol (Appendix A). However, Western blots were not useful for assessing the precise amount of HMGB1 that was present in the subcellular compartments. Therefore, nanoLC-PRM-HRMS was used to quantify the nuclear and cytosolic HMGB1 and compared with the HMGB1 that was secreted after treating the A549 cells with 50 µM cisplatin or 50 µM cisplatin with 4.5 nM LMB for 24 h (Figure 7). This revealed that there was a highly significant increase in nuclear HMGB1 from 2599 ± 182 ng/10^6^ cells in untreated cells to 3764 ± 237 ng/10^6^ cells (*p* < 0.0025, Figure 7A) after cisplatin treatment. Nuclear HMGB1 was further increased by 488 ng/10^6^ cells to 4272 ± 248 ng/10^6^ cells with a combination of cisplatin and LMB (Figure 7A). The high nuclear HMGB1 content after cisplatin treatment relative to the amount secreted (888 ± 33 ng/10^6^ cells, Figure 7C) meant that the increase did not quite meet the criteria for statistical significance with a *p* value of <0.0628). However, cytosolic HMGB1 was significantly lower (*p* < 0.0433) with a combination of cisplatin and LMB (2398 ± 268 ng/10^6^ cells) than untreated cells (3164 ± 367 ng/10^6^ cells), whereas cytosolic HMGB1 after treatment with cisplatin alone (2851 ± 846 ng/10^6^ cells) was not significantly different from untreated cells (Figure 7B). Conversely, there was a significant increase in the amount of HMGB1 secreted from the A549 cells from 206 ± 5 ng/10^6^ cells in untreated cells to 898 ± 13 ng/10^6^ cells after cisplatin treatment (*p* < 0.0001, Figure 7C). Secreted HMGB1 was reduced significantly to 437 ± 19 ng/10^6^ cells by a combination of cisplatin and LMB (*p* < 0.001, Figure 7C).

### 3.8. Effects of Platinum Drugs and DMSO on Lung Cancer Cell Viability

Since DMSO and cisplatin both induced extracellular release of HMGB1 (Figure 2B), we investigated whether in both cases its release resulted from passive diffusion through compromised nuclear and plasma membranes during A549 cell death. Following these treatments, we stained A549 cells with DNA stain Hoechst, pSIVA and necrosis-specific stain PI for imaging by fluorescence microscopy. We found that DMSO and cisplatin differed significantly in their effects on A549 viability and membrane integrity (Appendix A). When compared with the control (Appendix A), DMSO-treated cells demonstrated robust apoptosis and secondary necrosis, with significantly compromised integrity of nuclear and plasma membranes (Appendix A). Conversely, A549 cells treated with 20 µM cisplatin showed no signs of apoptosis or necrosis after 24 h (Appendix A), with necrotic cells beginning to appear with 100 µM cisplatin (Appendix A).

### 3.9. Carboplatin and Transplatin Are Less Effective than Cisplatin at Decreasing Cell Count and Viability

The decrease in the A549 cell count induced by 20 µM transplatin (6.6 × 10^6^ ± 1.5 × 10^5^, *p* < 0.0001), 100 µM transplatin (6.4 × 10^6^ ± 1.1 × 10^5^, *p* < 0.0001), 100 µM carboplatin (5.5 × 10^6^ ± 2.7 × 10^5^, *p* < 0.0001), and 300 µM carboplatin (4.7 × 10^6^ ± 1.8 × 10^5^, *p* < 0.0001), was significantly less than 20 µM cisplatin (3.8 ± 3.2 × 10^5^) (Appendix A). Similarly, the A549 cell viability induced by 20 µM transplatin (95.3 ± 1.0%, *p* < 0.0001), 100 µM transplatin (94.2 ± 0.9%, *p* < 0.0001), and 100 µM carboplatin (94.3 ± 0.6%, *p* < 0.0001) was also significantly lower than 20 µM cisplatin (91.4 ± 0.4%). However, there was no significant difference in cell viability between 20 µM cisplatin and 300 µM carboplatin (94.2 ± 2.1%).

### 3.10. Cisplatin and Leptomycin B (LMB) and Synergistically Attenuate A549 Cell Growth and Increase A549 Cell Death

The minimal amount of PI staining of cisplatin-treated cells indicated that the secretion of HMGB1 from A549 cells was not dependent on loss of nuclear and plasma membrane integrity (Appendix A). To further evaluate this result with an alternative approach, we analyzed cisplatin-treated A549 cells with trypan blue exclusion assays (Figure 8). These assays showed that 24 h treatment with 50 µM cisplatin primarily induced a decrease in cell proliferation (*p* < 0.001, Figure 8A upper) and viability (*p* < 0.001, Figure 8B upper) after 24 h when compared with PBS. There was a very modest increase in cell death (5.9 ± 1.4 × 10^5^ cells; *p* < 0.044) when compared with PBS (4.3 ± 0.5 × 10^5^ cells) at 24 (Figure 8C upper). A combination of 50 µM cisplatin with 16 nM leptomycin B (LMB) significantly increased cell death at 24 h to 7.4 ± 1.4 × 10^5^ cells when compared with 50 µM cisplatin alone (*p* < 0.0246) (Figure 8C, upper).

There remained a significant decrease in cell proliferation (*p* < 0.0001, Figure 8A lower) and viability (*p* < 0.0001, Figure 8B lower) after 50 µM cisplatin treatment when compared with PBS at 48 h with no additional increase in cell death (4.3 ± 0.9 × 10^5^; *p* < 0.042, Figure 8C lower). However, a combination of cisplatin and LMB resulted in a highly significant increase in cell death at 48 h of 10.2 ± 1.6 × 10^5^ cells when compared with LMB alone (*p* < 0.0001) or cisplatin alone (*p* < 0.0001) (Figure 8C, lower).

## 4. Discussion

Numerous quantitative studies of protein expression using MS-based methodology have relied on the utilization of stable isotopically labeled standards known as AQUA peptides [48]. These AQUA peptides offer exceptional precision by compensating for variations in ionization efficiency within the mass spectrometer. However, their accuracy is compromised, particularly when IP is employed for protein isolation, as they fail to account for losses that occur during the isolation procedure. Moreover, they do not consider inter-sample differences in protein digestion efficiency, as demonstrated in our research on apolipoprotein (Apo)A1 protein in human serum [49]. The inclusion of an AQUA peptide before protease digestion results in differential loss of the peptide during digestion when compared to the protein-derived peptide [49], rendering this approach unsuitable. Fortunately, these challenges can be effectively overcome by utilizing Stable Isotope Labeling by Amino Acids in Cell Culture (SILAC) protein internal standards, as we have successfully demonstrated for HMGB1 in serum and plasma [15], as well as in various other scenarios. For instance, SILAC internal standards have proven effective for quantifying oxidized HMGB1 in cell media [6], amyloid-β proteins in cerebrospinal fluid (CSF) [50], apolipoprotein-A1 in serum [51], human frataxin-M and human frataxin-E proteins in whole blood [52], and mouse frataxin-M proteins in mouse heart, brain, and liver tissues [47]. The SILAC protein internal standard method establishes a ratio between the SILAC-labeled protein and the corresponding endogenous protein at the beginning of the isolation procedure. This ratio remains constant throughout the entire process and is employed to calculate the quantity of endogenous protein using a standard curve constructed simultaneously with an authentic protein standard. Additionally, the SILAC protein serves as a carrier to enhance the recovery of low-level tissue proteins that may be lost due to non-selective binding to glassware and plastic surfaces. This effect was convincingly demonstrated in our assay for amyloid-β proteins in CSF [50], where most of the amyloid proteinsare lost in the absence of a stable isotope carrier through binding to surfaces during isolation and analysis. 

Immunoblotting of cell culture media with an anti-HMGB1 pAb demonstrated that REN and EMMESO mesothelioma cell lines constitutively secrete significant amounts of HMGB1. However, A549 NSLC cells do not secrete HMGB1 unless treated with cisplatin. In contrast, HMGB2, despite being found in similar amounts to HMGB1 in the nucleus of A549 cells, is not secreted. Absolute quantification with LC-MS on immunocaptured HMGB1 from A549 cell culture media revealed that cisplatin was very effective at inducing HMGB1 release (Figure 2B). This suggested that cisplatin has a unique biological activity, which causes HMGB1 secretion. In keeping with such a possibility, neither cisplatin’s stereoisomer transplatin nor the platinum drug analog carboplatin caused significant HMGB1 release, even at much higher doses. Both cisplatin [53] and carboplatin [54] form the same 1,2-(GpG) intrastrand crosslinks, suggesting that the slower kinetics of carboplatin’s reactivity with DNA might be sufficient to result in disparate effects on HMGB1 release, or the that activity might even be distinct from their effects on DNA damage. Trypan blue exclusion assays demonstrated that 100 µM carboplatin was less effective than 20 µM and 100 µM cisplatin in attenuating A549 cell proliferation and viability after 24 h, suggesting less impact on DNA replication, presumably through less reactivity with nuclear DNA (Appendix A). However, there was no difference in cell viability between 20 µM cisplatin and 300 µM carboplatin, which means that it is not simply a difference in the ability of platinum drugs to induce A549 cell death that is responsible for the increase in cisplatin-mediated HMGB1 release.

DMSO (*p* < 0.0014) and cisplatin at 20 µM (*p* < 0.0014) and 100 µM (*p* < 0.0001) induced significantly increased amounts of HMGB1 secretion from A549 cells when compared with PBS controls (Figure 2B). DMSO is known to cause cellular oxidative stress and DNA oxidation [43] as well as apoptosis [44]. Therefore, we reasoned that both DMSO and cisplatin might function similarly by causing the passive release of HMGB1 due to loss of nuclear and plasma membrane integrity following apoptosis and secondary necrosis. However, analysis with fluorescence microscopy revealed that DMSO caused robust apoptosis and necrosis (Appendix A), whereas low-dose cisplatin had very little effect after 24 h at doses up to 100 µM (Appendix A). Trypan blue exclusion assays confirmed this finding, showing that 24 h cisplatin treatment with 50 µM cisplatin primarily induced a significant decrease in cell proliferation (Figure 8A upper) and viability (Figure 8B upper) after 24 h compared to PBS. There was a very modest increase in cell death (5.9 ± 1.4 × 10^5^ cells; *p <* 0.044) when compared to PBS (4.3 ± 0.5 × 10^5^ cells) at 24 h (Figure 8C upper). Given that A549 cells showed significant secretion of HMGB1 with cisplatin despite intact nuclear and plasma membranes after 24-h, we reasoned that secretion was through a regulated mechanism rather than a passive release of nuclear proteins. It is noteworthy that there remained a significant decrease in cell proliferation (*p* < 0.0001, Figure 8A lower) and viability (*p* < 0.0001, Figure 8B lower) when compared with PBS at 48 h with no additional increase in cell death (Figure 8C lower).

To confirm that cisplatin treatment specifically induced the secretion of HMGB1, and not all nuclear proteins, we examined the nuclear, cytosolic, and media fractions of A549 cells for HMGB1 and HMGB2. Anti-HMGB1 and anti-HMGB1 immunoblotting showed specificity for HMGB1 and HMGB2, respectively, as demonstrated with recombinant proteins (Appendix A). HMGB1 was detected in nuclear and cytosolic fractions (Appendix A), whereas HMGB2 was only detected in the nucleus. (Appendix A). Anti-histone 4 and anti-FASN immunoblotting confirmed robust enrichment of the nuclear and cytosolic fractions and showed that there was no leakage of A549 cellular proteins into the cell culture media (Appendix A). HMGB1 and HMGB2 are nuclear proteins that are 93% homologous and 80% identical [55]. Surprisingly, we found that only HMGB1 was secreted in significant amounts into the A549 cell culture media after cisplatin treatment. This finding was confirmed by qualitative LC-MS analysis of HMGB1- and HMGB2-specific peptides in subcellular fractions (Appendix A). It has been previously reported that HMGB2 can be secreted from myeloid cells [56] and has been detected in serum of patients with coronary artery in-stent restenosis [57], which provides further evidence that cisplatin is unique in its ability to selectively induce HMGB1 release in the presence of high levels of HMGB2.

There are several reports that acetylation of HMGB1 at K-3, K-7, or K-12 can modulate its ability to inhibit the repair of cisplatin DNA adducts [4,58,59,60]. Unfortunately, in vitro, and in vivo studies with cisplatin have been unable to confirm that these PTMs are present on secreted HMGB1. To address this issue by quantifying acetylation at K-3, K-7, and K-12, a SILAC HMGB1 internal standard was added to cell culture media samples collected from A549 cells (Figure 1). HMGB1 was isolated by IP of these samples and treated with *D*6-acetic anhydride to trideuteroacetylate (*CD*_3_-acetylate) all unacetylated HMGB1 lysine residues. This process converted each of the HMGB1-derived peptides into one molecular form that only differed in the mass of the acetyl-lysine modification. In a previous study, we showed the IP procedure can isolate HMGB1 containing acetylated lysine residues [15]. Using this approach, we were unable to detect acetylation on K-3, K-7, or K-12 residues using a highly specific nanoLC-PRM/HRMS method (Figure 1). There is also substantial evidence suggesting that HMGB1 NLS1 and NLS2 acetylation is required to facilitate its secretion [31,32,41,61], although many of the key studies have been retracted or statements of concern issued [28]. Consequently, it is perhaps not surprising that we were unable to detect any acetylated forms of HMGB1 secreted after treatment of A549 lung cells with cisplatin. Therefore, we investigated whether cisplatin-mediated HMGB1 secretion was regulated by other PTMs.

The oxidation state of HMGB1′s three Cys residues has been considered especially important in regulating both HMGB1′s secretion and its extracellular signaling activity. However, there are contradictory reports that relate the redox status of extracellular HMGB1 to its mechanism of secretion. This is because most approaches in interrogating the redox state of HMGB1 lack the necessary quantitative parameters for site-specific analysis to accurately measure labile modifications like disulfide bonds. Therefore, we developed a novel differential alkylation approach using SILAC HMGB1 internal standard to control for any differences in HMGB1 recovery, digestion, or peptide ionization efficiency among differentially alkylated HMGB1 proteoforms. This made it possible to conduct site-specific quantification of HMGB1 Cys-disulfides (Figure 2). Although other tested treatments induced the release of some reduced HMGB1 with free Cys-23 and C-45, cisplatin induced the secretion of significantly more reduced HMGB1. The association of this reduced proteoform of HMGB1 with chemokine activity [62,63,64] indicates that cisplatin mediated HMGB1 secretion could activate immune cells during cancer chemotherapy.

We also explored whether other PTMs might be involved in the regulation of cisplatin-mediated HMGB1 secretion by testing the effects of different pharmacological inhibitors of PTM formation and examining immunoblots of the cell culture media. Although ADP-ribosylation by PARP1 has been implicated in the regulation of HMGB1 mobility, including a model of DNA damage-associated subcellular translocation [65], the PARP1 inhibitor Olaparib [66] had no effect on cisplatin-mediated HMGB1 secretion (Figure 6A). Similarly, although HMGB1 phosphorylation by PKCs has been suggested to increase HMGB1 secretion, we found that pan-PKC inhibitors GO-6983 [67] and Rottlerin [68] had no effect on cisplatin-mediated HMGB1 secretion from A549 cells (Figure 6B,C). Furthermore, we were unable to detect any other modifications with lysine, tyrosine, serine, or threonine using LC-HRMS/MS analysis of Asp-N, Glu-C, or chymotryptic peptides.

Misinterpretation of much of the original MS data on HMGB1 secretion led to the erroneous concept that acetylation was required for HMGB1 secretion to occur [69]. Nevertheless, this early study did convincingly show that inhibition of nuclear export with LMB prevented the transport of HMGB1 from the nucleus to the cytosol [69]. However, this study did not report that LMB also inhibits secretion of the substantial amounts of HMGB1 that are present in the cytosol (Figure 7B). LMB is known to inhibit the nuclear export of proteins through binding and inhibition of XPO1 (CRM1) (Figure 9) [70,71]. Inhibition of XPO1 might also prevent the nuclear export of pro-secretory proteins and so prevent the secretion of cytosolic proteins [72]. We reasoned that LMB could inhibit the secretion of HMGB1 by concentrating it in the nucleus and inhibiting secretion of the residual HMGB1 from the cytosol (Figure 9). LMB caused an increase in cell death (Figure 8C) and is known to be cytotoxic [73]. 

Consequently, there is a narrow concentration range where a decrease in HMGB1 secretion can be observed before it induces necrotic cell death and releases HMGB1. We found that the addition of LMB caused a significant inhibition of cisplatin-mediated HMGB1 secretion (Figure 6D,F and Figure 7C). We also showed that the XPO1 inhibitor KPT-330 [74,75] also caused a significant decrease in HMGB1 secretion (Figure 6E,G). A combination of cisplatin and LMB resulted in an additive increase in A549 cell death at 24 h and 48 h when compared with LMB alone or cisplatin alone (Figure 8C). There was a concomitant increase in nuclear HMGB1 with a decrease in cytosolic HMGB1 (Figure 7B) and inhibition of HMGB1 secretion (Figure 7C) as would be predicted if nuclear export were inhibited (Figure 9). HMGB1 binds more tightly to cisplatinated DNA than unmodified DNA, and this binding inhibits the repair of cisplatin-derived 1,2-(GpG) DNA intrastrand crosslinks, which in turn interferes with DNA replication and eventually activates apoptosis [25]. Cisplatin treatment causes increased HMGB1 concentration in the nucleus (Figure 7A), and cancer cells seem to partially survive treatment with cisplatin by transporting HMGB1 from the nucleus into the cytosol where it is then secreted into the extracellular milieu. Therefore, our data are consistent with the concept that LMB-mediated accumulation of HMGB1 in the nucleus facilitates the known ability of HMGB1 to inhibit 1,2-(GpG) DNA intrastrand repair [4,58,59,60] and so potentiates the ability of cisplatin to induce A549 NSCLC viability and cell death (Figure 7).

## 5. Conclusions

We developed a stable isotope dilution nano-LC-PRM/HRMS assay for the precise and accurate quantification of nuclear, cytosolic, and secreted HMGB1 proteoforms. This revealed that inhibition of nuclear transport (Figure 9) inhibits cisplatin-mediated HMGB1 secretion (Figure 6) and causes HMGB1 accumulation in the nucleus (Figure 7A), which is known to inhibit of the repair of 1,2-(GpG) intrastrand crosslinks [4,58,59,60]. Inhibition of DNA repair by the increased nuclear HMGB1 was most likely responsible for the decreased cell viability (Figure 8B) and increased cell death that was observed (Figure 8C) [25]. This offers the potential for a novel approach to increasing the antitumor activity of low-dose cisplatin by co-administration with low doses of XPO1 inhibitors such as KPT-330 (Selinexor). The difference between A549 NSCLC cells and malignant mesothelioma REN and EMMESO cells regarding HMGB1 secretion (Figure 2A) suggests that different cancer cells will differ in their capacity to secrete HMGB1 in response to cisplatin therapy. The findings of this research have established a different mechanism of HMGB1 secretion in a NSCLC model than has previously been reported. Therefore, future research will benefit from examining this new mechanism in additional cancer cell lines.

It is intriguing that the nano-LC-PRM/HRMS assay also revealed that cisplatin but not carboplatin stimulated HMGB1 secretion from A549 NSCLC cells (Figure 2) as these chemotherapeutic agents are typically thought to have identical modes of action. HMGB1 is an immunomodulator through activation of RAGE, TLR2, and TLR4 [7,8]. The immunomodulatory activity of HMGB1 arises primarily through activity of the fully reduced form, a major proteoform secreted from A549 cells by treatment with doses of cisplatin that slow cell growth but cause minimal cell death after 24 h (<100 µM). Finally, there is increasing interest in combining low-dose (metronomic) chemotherapy in combination with immunotherapy for treating NSCLC [76,77]. Consequently, low-dose cisplatin-mediated HMGB1 release could be used to stimulate the immune system prior to treatment of NSCLC patients with immunotherapy. Furthermore, optimization of dose regimens could be informed by monitoring HMGB1 proteoform levels in the plasma using the methodology that we have reported previously [15].

## Data Availability

The data presented in this study are available on request from Ian A. Blair.

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
