# Peer review of "Cisplatin Dependent Secretion of Immunomodulatory High Mobility Group Box 1 (HMGB1) Protein from Lung Cancer Cells"

_biomolecules, 2023, doi:10.3390/biom13091335_

Round 1

Reviewer 1 Report

The manuscript submitted by Gillespie et al. examines the cellular translocation and export of HMGB1 following cisplatin treatment. The authors describe a novel MS method for quantifying HMGB1 proteins which is interesting and of potential utility in future studies. However, aside from this new methodology, the novelty is limited and there are a number of issues with the scientific approach and conclusions drawn from the presented data. The paper cited by the authors from Bonaldi et al. has previously shown HMGB1 is shuttled from nucleus to cytosol in an XPO1 dependent fashion, limiting the novelty here. The experimental approach for several assays is questionable and not convincing, in particular the cell death assays, which could drastically affect interpretation of results. The authors also suggest cisplatin leads to reduction in HMGB1 based on data in Figure 5. However, this data is discordant depending on whether heavy or light peptides are assessed (i.e., heavy peptides are more abundant under reducing conditions; light peptides more abundant with oxidizing conditions). Additionally, the H:L ratio is inconsistent across peptides in all presented MS data (in spite of contrary claims in lines 718-720) calling precise quantification into question. Overall the study has potential but there are a number of items that need to be addressed prior to publication.

Author Response

The manuscript submitted by Gillespie et al. examines the cellular translocation and export of HMGB1 following cisplatin treatment. The authors describe a novel MS method for quantifying HMGB1 proteins which is interesting and of potential utility in future studies. However, aside from this new methodology, the novelty is limited and there are a number of issues with the scientific approach and conclusions drawn from the presented data.

Answer: We appreciate the positive comments about novelty and potential utility of our new method and have fully addressed the concerns below:

  1. The paper cited by the authors from Bonaldi et al. has previously shown HMGB1 is shuttled from nucleus to cytosol in an XPO1 dependent fashion, limiting the novelty here.

Answer 1: The Bonaldi paper did not show that HMGB1 was secreted from cells (Ref 66).  In contrast, we have convincingly demonstrated that HMGB1 is secreted by cisplatin from NSCLC by rigorously quantifying the amount of secreted HMGB1 (Figure 2).  We used a modification of the mass spectrometry procedure procedure that we reported in previously (Ref 16).  We also definitively established the oxidation state of HMGB1 by making derivative of the reduced form of HMGB1 followed by a reduction step and formation of a different derivative to quantify the amount of oxidized HMGB1 that was present originally (Scheme 2).  Therefore, this is first report that inhibition of nuclear transport can also inhibit cisplatin mediated secretion of HMGB1, an important immunomodulatory molecule.  We also definitively established, for the first time, the oxidation state of secreted HMGB1 (Figure 4).  We hope that reviewer will consider that our findings are both important and novel.  The text has been modified to highlight the novelty of our finding as follows:

Lines 896-901

"Misinterpretation of much of the original MS data on HMGB1 secretion led to the erroneous concept that acetylation was required for HMGB1 secretion to occur [66].  Nevertheless, this early study did convincingly show that inhibition of nuclear export with LMB prevented the transport of HMGB1 from the nucleus to the cytosol [66].  However, this study did not report that LMB also inhibits secretion of the substantial amounts of HMGB1 that are present in the cytosol (Figure 7B)."

  1. The experimental approach for several assays is questionable and not convincing, in particular the cell death assays, which could drastically affect interpretation of results.

Answer 2: The trypan blue exclusion assay is a widely used and robust to determine the number of viable cells present in a cell suspension.  It is based on the principle that live cells possess intact cell membranes that exclude certain dyes, such as trypan blue, eosin, or propidium, whereas dead cells do not.  Cells are suspended in  PBS containing trypan blue and then examined to determine the percentage of cells that have a clear cytoplasm (viable cells) versus cells that have blue cytoplasm (nonviable dead cells).  We realize that there is a substantial amount of work required to specifically characterize the type of cell death induced by cisplatin, but this was not the main purpose of the present study.  The most likely interpretation of our data is that inhibition of the nuclear exporter XPO1 results in reduced secretion of HMGB1 (Figure 7C) the accumulation of HMGB1 in the nucleus (Figure 7A), which reduces cell viability (Figure 8B) and increases cell death (Figure 8C).

  1. The authors also suggest cisplatin leads to reduction in HMGB1 based on data in Figure 5. However, this data is discordant depending on whether heavy or light peptides are assessed (i.e., heavy peptides are more abundant under reducing conditions; light peptides more abundant with oxidizing conditions). Additionally, the H:L ratio is inconsistent across peptides in all presented MS data (in spite of contrary claims in lines 718-720) calling precise quantification into question.

Answer 3: The stable isotopically labeled protein (SILAC-HMGB1) was added to the biological matrix before protein isolation, cysteine derivatization, reduction, protease digestion to generate the peptides, and mass spectrometry-based analysis of the peptides.  The stable isotopically labeled (heavy) peptides that are generated by the protease digestion have nothing to do with the biology.  Therefore, their abundance has nothing to do with reducing or oxidizing conditions.  They simply serve as internal controls that permit protein concentrations to be determined with precision and accuracy in the range of ± 15 %.  We have previously used SILAC labeled internal standards to achieve this  precision and accuracy for HMGB1 (Ref 15) as well as other proteins (e.g. refs 6, 47, 50-52).  Our methods have been rigorously validated and are not questionable.  In fact, there is no other way to analyze proteins present in cells and biofluids with such accuracy and precision as we described in detail in the original manuscript (new lines 763 to 779):

“Fortunately, these challenges can be effectively overcome by utilizing Stable Isotope Labeling by Amino Acids in Cell Culture (SILAC) protein internal standards, as we have successfully demonstrated for HMGB1 in serum and plasma [15], as well as in various other scenarios.  For instance, SILAC internal standards have proven effective for quantifying oxidized HMGB1 in cell media [6], amyloid-β proteins in cerebrospinal fluid (CSF) [50], apolipoprotein-A1 in serum [51], human frataxin-M and human frataxin-E proteins in whole blood [52], and mouse frataxin-M proteins in mouse heart, brain, and liver tissues [47].  The SILAC protein internal standard method establishes a ratio be-tween the SILAC-labeled protein and the corresponding endogenous protein at the beginning of the isolation procedure.  This ratio remains constant throughout the entire process and is employed to calculate the quantity of endogenous protein using a standard curve constructed simultaneously with an authentic protein standard.  Additionally, the SILAC protein serves as a carrier to enhance the recovery of low-level tissue proteins that may be lost due to non-selective binding to glassware and plastic surfaces.  This effect was convincingly demonstrated in our assay for amyloid-β proteins in CSF [50], where proteins are nearly entirely lost in the absence of a stable isotope carrier due to binding to surfaces during isolation and analysis.”

  1. Overall the study has potential but there are a number of items that need to be addressed prior to publication.

Answer 4: We really appreciate the reviewer’s positive comment and hope that we have satisfactorily addressed concerns about the novelty (answer 1), the cell death assay (answer 2), and the protein quantification (answer 3), so that the manuscript is now acceptable for publication.

Reviewer 2 Report

This manuscript revealed that cisplatin, but not carboplatin or transplatin, could induce HMGB1 release from A549 cells prior to cell death or loss of plasma membrane integrity. Furthermore, the secretion was mediated by nuclear transport rather than by acetylation, ADP-ribosylation, phosphorylation, or oxidation.These findings suggest that inhibition of XPO1 could potentiate the anti-tumor activity of cisplatin by increasing the nuclear accumulation of HMGB1 protein. Furthermore, low-dose cisplatin therapy could modulate the immune response in NSCLC through the established chemokine activity of extracellular reduced HMGB1. This could potentially enhance the efficacy of subsequent immunotherapy treatment. It is a sound story. But I have several following concerns:

1) In Line 39, "2 and 4"should be "-2 and-4".

2) Abbreviations should be defined When they appear first time. Such as "PI", "MS", "SDS-PAGE"...

3) "p" denoting a significant difference should be italicized.

4) In Figure 2, why did the authors measure HMGB1 production in cell culture supernatants using WB instead of the more commonly used and convenient ELISA assay?

5) Please agree on the format of references in the text, including the name of the author, the case of the words in the title, and the page number.

Minor editing of English language required.

Author Response

This manuscript revealed that cisplatin, but not carboplatin or transplatin, could induce HMGB1 release from A549 cells prior to cell death or loss of plasma membrane integrity. Furthermore, the secretion was mediated by nuclear transport rather than by acetylation, ADP-ribosylation, phosphorylation, or oxidation. These findings suggest that inhibition of XPO1 could potentiate the anti-tumor activity of cisplatin by increasing the nuclear accumulation of HMGB1 protein. Furthermore, low-dose cisplatin therapy could modulate the immune response in NSCLC through the established chemokine activity of extracellular reduced HMGB1. This could potentially enhance the efficacy of subsequent immunotherapy treatment. It is a sound story.

Answer: We really appreciate these positive comments.

But I have several following concerns:

1, In Line 39, "2 and 4"should be "-2 and-4".

Answer 1: This has been corrected (new line 40).

2, Abbreviations should be defined When they appear first time. Such as "PI", "MS", "SDS-PAGE"...

Answer 2: Abbreviations for PI, MS, pSIVA, SDS-PAGE, ADP, and DMSO have been defined.

3, "p" denoting a significant difference should be italicized.

Answer 3: This has been corrected throughout.

  1. In Figure 2, why did the authors measure HMGB1 production in cell culture supernatants using WB instead of the more commonly used and convenient ELISA assay?

Answer 4: In Figure 2A, HMGB1 was quantified by stable isotope dilution nanoLC-PRM/HRMS, which we definitively showed was far more accurate than ELISA assays for HMGB1 in our 2018 Analytical Chemistry paper (Ref 15).  ELISA methods for HMGB1, which do not use an internal control, consistently over-estimate the amount of HMGB1 and/or do not discriminate between HMGB1 and HMGB2 as discussed in detail in Ref 15. The Western blot in Figure 2A was simply provided as a qualitative comparison of cell culture supernatants and was not used to generate the quantitative analysis depicted in Figure 2B.

  1. Please agree on the format of references in the text, including the name of the author, the case of the words in the title, and the page number.

Answer 5. The references were formatted according to the current version of End Note supplied by Biomolecules.  We will leave it to the editorial staff to modify according to the latest Biomolecules format.

Comments on the Quality of English Language

Minor editing of English language required.

Answer: The manuscript has been thoroughly edited.

Reviewer 3 Report

1. The statement in P.20 lines 803-4: “Most approaches to interrogating HMGB1 oxidation lack the necessary quantitative parameters …” should be corrected as “Most approaches in interrogating HMGB1 oxidation lack the necessary quantitative parameters …”.

2. The authors should cite some references to support the statement in P.21 lines 811-13: “The association of this reduced proteoform of HMGB1 with chemokine activity indicates that cisplatin could increase immune cell recruitment to tumors during cancer chemotherapy”.

3. The statement in P.22 lines 848-50: “A combination of cisplatin and LMB resulted in a synergistic increase in A549 cell death at 24-h and 48-h when compared with LMB alone or cisplatin alone (Figure 8C)” is incorrect because the increase was “additive” at best.

4. The authors should provide evidence to support their idea that LMB potentiates the ability of cisplatin to induce A549 cell death by increasing the accumulation of HMGB1 in the nucleus which facilitates the inhibition of 1,2-(GpG) DNA intrastrand repair (P.22 lines 859-62).

Minor English editing is required.

Author Response

  1. The statement in P.20 lines 803-4: “Most approaches to interrogating HMGB1 oxidation lack the necessary quantitative parameters …” should be corrected as “Most approaches in interrogating HMGB1 oxidation lack the necessary quantitative parameters …”.

Answer 1: The text has been modified as follows:

Lines 855-860

The oxidation state of HMGB1’s three Cys residues has been considered especially important in regulating both HMGB1’s secretion and its extracellular signaling activity.  However, there are contradictory reports that relate the redox status of extracellular HMGB1 to its mechanism of secretion.  This is because most approaches in interrogating the redox state of HMGB1 lack the necessary quantitative parameters for site-specific analysis to accurately measure labile modifications like disulfide bonds.

  1. The authors should cite some references to support the statement in P.21 lines 811-13.

Answer 2: The text has been modified as follows:

Lines 866-884

“The association of this reduced proteoform of HMGB1 with chemokine activity [62-64] indicates that cisplatin could increase immune cell recruitment to tumors during cancer chemotherapy.

New references

  1. Venereau, E.; Casalgrandi, M.; Schiraldi, M.; Antoine, D.J.; Cattaneo, A.; De Marchis, F.; Liu, J.; Antonelli, A.; Preti, A.; Raeli, L.; et al. Mutually exclusive redox forms of HMGB1 promote cell recruitment or proinflammatory cytokine release. J Exp Med 2012, 209, 1519-1528, doi:10.1084/jem.20120189.
  2. Schiraldi, M.; Raucci, A.; Munoz, L.M.; Livoti, E.; Celona, B.; Venereau, E.; Apuzzo, T.; De Marchis, F.; Pedotti, M.; Bachi, A.; et al. HMGB1 promotes recruitment of inflammatory cells to damaged tissues by forming a complex with CXCL12 and signaling via CXCR4. J Exp Med 2012, 209, 551-563, doi:10.1084/jem.20111739.
  3. Venereau, E.; Schiraldi, M.; Uguccioni, M.; Bianchi, M.E. HMGB1 and leukocyte migration during trauma and sterile inflammation. Mol Immunol 2013, 55, 76-82, doi:10.1016/j.molimm.2012.10.037.

3. The statement in P.22 lines 848-50: “A combination of cisplatin and LMB resulted in a synergistic increase in A549 cell death at 24-h and 48-h when compared with LMB alone or cisplatin alone (Figure 8C)” is incorrect because the increase was “additive” at best.

Answer 3: The text has been modified as follows:

Lines 925-926

“LMB resulted in an additive increase in A549 cell death at 24-h and 48-h when compared with LMB alone or cisplatin alone (Figure 8C).”

  1. The authors should provide evidence to support their idea that LMB potentiates the ability of cisplatin to induce A549 cell death by increasing the accumulation of HMGB1 in the nucleus which facilitates the inhibition of 1,2-(GpG) DNA intrastrand repair (P.22 lines 859-62).

Answer 4: We showed that LMB increases the accumulation of HMGB1 in the nucleus.  HMGB1 is known to inhibit 1,2-(GpG) DNA intrastrand repair.  Therefore, it is logical to suggest that LMB-mediated accumulation of HMGB1 in the nucleus would cause increased inhibition of 1,2-(GpG) DNA intrastrand repair.  As far as we are aware, there are no reports that increased HMGB1 concentrations do not cause an increase in the inhibition of 1,2-(GpG) DNA intrastrand repair.

The text has been modified as follows:

Lines 935-938

“Therefore, our data are consistent with the concept that LMB-mediated accumulation of HMGB1 in the nucleus facilitates the known ability of HMGB1 to inhibit of 1,2-(GpG) DNA intrastrand repair [4, 58-60] and so potentiates the ability of cisplatin to induce A549 NSCLC viability and cell death (Figure 7).”

  1. Comments on the Quality of English Language

Minor English editing is required.

Answer 5.  The text has been thoroughly edited.

Round 2

Reviewer 2 Report

The authors have addressed all my concerns. I recommend accepting this manuscript in current status.

Author Response

Thanks.